# The Advent of Domain Adaptation into Artificial Intelligence for Gastrointestinal Endoscopy and Medical Imaging

**DOI:** 10.3390/diagnostics13193023

**Published:** 2023-09-22

**Authors:** Min Ji Kim, Sang Hoon Kim, Suk Min Kim, Ji Hyung Nam, Young Bae Hwang, Yun Jeong Lim

**Affiliations:** 1Division of Gastroenterology, Department of Internal Medicine, Dongguk University Ilsan Hospital, Dongguk University College of Medicine, Goyang 10326, Republic of Korea; immj2020@naver.com (M.J.K.); spring0107@naver.com (S.H.K.); drnamesl@dumc.or.kr (J.H.N.); 2Department of Intelligent Systems and Robotics, College of Electrical & Computer Engineering, Chungbuk National University, Cheongju 28644, Republic of Korea; zhdnrk1@naver.com (S.M.K.); ybhwang@cbnu.ac.kr (Y.B.H.)

**Keywords:** domain adaptation, endoscopy, artificial intelligence, CycleGAN

## Abstract

Artificial intelligence (AI) is a subfield of computer science that aims to implement computer systems that perform tasks that generally require human learning, reasoning, and perceptual abilities. AI is widely used in the medical field. The interpretation of medical images requires considerable effort, time, and skill. AI-aided interpretations, such as automated abnormal lesion detection and image classification, are promising areas of AI. However, when images with different characteristics are extracted, depending on the manufacturer and imaging environment, a so-called domain shift problem occurs in which the developed AI has a poor versatility. Domain adaptation is used to address this problem. Domain adaptation is a tool that generates a newly converted image which is suitable for other domains. It has also shown promise in reducing the differences in appearance among the images collected from different devices. Domain adaptation is expected to improve the reading accuracy of AI for heterogeneous image distributions in gastrointestinal (GI) endoscopy and medical image analyses. In this paper, we review the history and basic characteristics of domain shift and domain adaptation. We also address their use in gastrointestinal endoscopy and the medical field more generally through published examples, perspectives, and future directions.

## 1. Introduction

Artificial intelligence (AI) has attracted significant attention in medical image analyses. Gastrointestinal (GI) endoscopy is an active area of AI research. Upper endoscopies, colonoscopies, and capsule endoscopies detect inflammation, bleeding foci, preneoplastic lesions, and gastrointestinal cancer. The goals of AI research are to improve our ability to detect abnormal lesions, as well as enhance gastrointestinal imaging and its quality control and clinical efficiency in real clinical environments. The applications of AI in GI endoscopy range from computer-aided detection (CAD) to objective assessments of the degree of bowel preparation. In particular, CAD can potentially reduce endoscopic reading time and dramatically increase accuracy. Convolutional neural networks (CNNs) are primary deep learning algorithms for endoscopic image processing. CNN-based algorithms have been successful in detecting a variety of esophageal, stomach, small bowel, and colorectal images using different imaging modalities [1,2,3,4,5,6].

At first glance, it appears that the implementation of an automated reading system for GI endoscopy should be a simple task. However, there are some major obstacles to it reaching the clinical level. The endoscopic images collected for the training and testing of the AI are not homogeneous. More specifically, there are various endoscopy manufacturers worldwide. In addition, even among products from the same manufacturer, the image characteristics differ depending on the filming environment or image-processing software. Therefore, the development of a universal algorithm that is also highly versatile is a challenge. When an AI algorithm trained with images obtained using a device from Company A is applied to images obtained using a device from Company B or another manufacturer, there is a concern that the accuracy of the AI may drop significantly. This is called a “domain shift problem”. It has already been reported that this domain shift problem occurs even when an endoscopic image from one hospital is applied to an AI module adapted in another hospital.

Domain adaptation is a promising tool for overcoming the problem of multimodal endoscopic image acquisition for AI development. It involves feature alignment, an image alignment process that typically uses a framework for image conversion to reduce the differences in appearance among images [7,8]. Domain adaptation technology forms the basis for creating a universal AI system for endoscopic image analyses.

This review encompasses the fundamentals and history of domain adaptation, its applications in GI endoscopy and the medical field more generally, and future perspectives.

## 2. Fundamentals and History of Domain Adaptation and Domain Shift

Domain adaptation began to be studied when generative adversarial networks (GAN), now one of the most widely used generative models, were introduced [9]. Previously, it was difficult to apply transformations between various domains, so it was difficult to generate a proper dataset. In a CycleGAN paper published in 2017, the concept of cycle consistency loss was introduced. This stated that the same image should be generated when converting from the source domain to the target domain, and then from the target domain to the source domain, even if there is no exact correspondence. Many studies on domain adaptation have now been conducted.

To implement domain adaptation, it is first necessary to understand deep-learning-based features, such as CNNs, which extract the important information from images. A GAN can be trained by understanding the different loss functions compared to those used in the existing CNN. It should be noted that the generator and discriminator are used simultaneously during training; however, only the generator is used to create an image for the new domain after learning. To implement this, the Python programming language is used, and open libraries for deep learning, such as TensorFlow or PyTorch, are installed [10,11,12].

GANs have had a significant impact on the field of AI by transforming supervised-learning-oriented deep learning systems into unsupervised learning. Previously, learning was possible only when all data were correct. However, the emergence of GANs, which can continuously generate new data without correct answers, has opened up new possibilities for AI; these include image generation and restoration, domain transformation, object detection, super resolution, and music generation. Domain adaptation refers to the adaptation of information from existing domains to new domains that are different but relevant. Here, the two main domains are divided into source and target domains and are assumed to have different dataset distributions (Figure 1) [7]. If the model learned from the source domain is to be applied to the target domain, a problem inevitably arises because the two domains have different distributions, which are usually referred to as “domain shifts”. Therefore, the goal of domain adaptation is to create a more robust model for these domains to prevent domain shifts.

Such a model can be leveraged to obtain a greater accuracy by increasing the amount of insufficient learning data, and existing learned networks can be exploited through domain translation without further learning. Because these domain conversion techniques can be applied to various existing object recognition problems, they have been used in a number of recently conducted studies [13]. GAN studies for domain transformation are divided into non-style code models such as Pix2Pix, CycleGAN, and DiscoGAN, and style code models such as MUNIT, DRIT, and StarGAN [14,15]. Depending on the structure of the model, the same data can produce different results. Therefore, it is important to use an appropriate network based on the learning data [16,17].

A generative adversarial network, or GAN, is a network in which constructors and identifiers compete with each other and generate data. The goal of a GAN is to generate data close to the distribution of the real data, and the generator attempts to generate fake data close to the real data so that the discriminator does not falsely discriminate (Figure 2). The goal of this process is to gradually improve the performances of the generator and discriminator and ultimately prevent the discriminator from distinguishing between real and fake data [18].

Pons et al. determined the performance before and after domain adaptation using a receiver operating characteristic curve (Figure 3) [19]. The performance of the baseline image before domain adaptation was extremely poor (area under the curve (AUC), 0.6488). After the domain adaptation with CycleGAN, the true positive rate of the image was significantly improved (AUC, 0.7341).

## 3. Application of Domain Adaptation GI Endoscopy and Medical Field

Table 1 summarizes the many studies that have been conducted on domain adaptation in the medical field.

### 3.1. Using Triplet Loss for Domain Adaptation in Wireless Capsule Endoscopy (WCE)

Since the announcement of the first WCE device in 2001, technological advances have progressed rapidly; new devices have been regularly introduced, offering increasingly good performances, better image resolution, better illumination, and larger fields of view. Today, WCE devices are produced by different manufacturers and offer a range of technical specifications [20,21].

We recently conducted a domain adaptation study using WCE. The latest AI models for automated WCE reading are useful with respect to reductions in reading time [22,23,24]. In addition, we used AI for automated abnormal lesion detection in small bowel WCE and found that the performance of the AI-assisted interpretation was comparable with that of experienced endoscopists for abnormal lesion detection without improvement. However, the performance of the optimized WCE AI algorithm deteriorated when it was applied to data from other hospitals. In addition, our previous study showed that a binary classification model (two categories of images: clinically insignificant images, including normal mucosa, bile, air bubbles, and debris; and significant lesion images, including inflammation, abnormal vascular lesions, and bleeding) produced excellent outcomes in an internal hospital test, with a high sensitivity even in unseen images. Unfortunately, this model showed subclinical outcomes in an external test at a third-party hospital. This phenomenon is also commonly experienced in AI applications to other images in real-world situations. This constitutes a major obstacle to the universal use of AI in medical fields. The domain shift problem has been also observed in WCE. AI engineers have adopted the domain adaptation technique to enable efficient algorithm learning by creating new virtual data and extracting the main features of a collected image. An A-image and a B-image are handled separately and divided into training and validation (test) sets. Two independent AI algorithms (A-AI and B-AI) are built using the training images from the two groups. These are trained to discriminate between normal and significantly abnormal images. After the training, each AI algorithm is applied to a homogeneous test set. These algorithms are then tested using a heterogeneous test set (e.g., applying A-AI to the B-image test set, or applying B-AI to the A-image test set) to determine whether the performance improves or deteriorates. Their performances are evaluated using heterogeneous test images after several up-to-date domain adaptation techniques are applied to the test images. Thus, we attempted to evaluate the extent to which the domain adaptation technique can improve the reading accuracy in heterogeneous images (Figure 4). We tested three different domain adaptation architectures, introduced from 2017 to present: CycleGAN, discoGAN, and MUNIT. These domain adaptation methods enable the original images of WCE to be transformed and reconstructed into a new image to match the distribution of the target domain. This process allows the newly created image to possess a style similar to the heterogenous capsule image. We also compared various domain adaptation techniques for an EfficientNet-based algorithm and ResNet-based algorithm. We found that the domain adaptation was effective regardless of the algorithm used. In addition, domain adaptation using CycleGAN resulted in an additional AI performance improvement. A similar improvement in terms of performance improvement was obtained using discoGAN. The improvement in terms of efficacy outcome was the greatest when the domain adaptation was performed with CycleGAN.

It is not surprising that, if a model is trained with data from an older capsule, it may not yield the expected results when it is evaluated with a newer capsule, because the same distribution of data is not guaranteed. However, it is inefficient and expensive to abandon an old database and create a new one from scratch each time a new device is developed. To overcome this problem, the authors propose a domain adaptation method based on deep metric learning using triplet loss. The aim of this method is to adapt an embedding space trained with a large training dataset to a new domain, in which comparatively few labeled images are available. The embedding space is adapted by generating triplets of images from both domains with the goal of two images in the same category being closer than images belonging to different domains. The research results show that, by using a small, labeled dataset from the new domain, the embedding space can perform well. Effective results may be readily obtained in the new environment using domain adaptation with only a few labeled images from older camera systems [25].

### 3.2. Colonoscopy Polyp Detection: Domain Adaptation from Medical Report Images to Real-Time Videos

Manually annotating polyp regions in large-scale video datasets is time-consuming and expensive, and has thus limited the development of deep learning techniques. To compensate for this, researchers have used labeled images to train target models and infer colonoscopy videos. However, there are many problems with image-based training and video-based inference, including domain differences, a lack of positive samples, and temporal smoothness. To address these issues, the use of an image-video-joint polyp detection network (Ivy-Net), which is a type of domain adaptation, has been proposed to address the domain gap between colonoscopy images from historical medical reports and real-time videos. In Ivy-Net, a modified mix-up is utilized to generate training data by combining positive and negative video frames at the pixel level, which can learn domain-adaptive representations and augment positive samples. Experiments on collected datasets have demonstrated that Ivy-Net achieves state-of-the-art results and significantly improves the average precision of the polyp detection in colonoscopy videos [26,27,28].

### 3.3. Unsupervised Adversarial Domain Adaptation for Barrett’s Segmentation

Because Barrett’s esophagus (BE) is a precancerous lesion, it is important to accurately identify the BE region, so that patients may be adequately monitored and minimally invasive therapy may be administered. Automated segmentation using AI helps clinical endoscopists to evaluate this BE area more accurately and thus determine a range of treatments [29,30]. The existing automated segmentation methods use CNN-based supervised models [31]. These supervised models require a large number of manual annotations that incorporate all data variability into the AI training data, and these fully supervised models often cannot be generalized to different imaging modalities because of domain shifts. This problem can be alleviated by applying unsupervised domain adaptation (UDA). UDA is trained on white-light images as the source domain, and is well-adapted for generalization to produce segmentation on different imaging modalities as the target domain; these include narrow-band imaging (NBI) and post-acetic-acid (PAA) white-light imaging. This approach has been found to provide a generalized prediction of the segmentation masks of unlabeled endoscopy images in cross-modalities and improve performance with respect to both NBI and PAA images. This method does not rely on the existence of target labels and provides accurate and stable results under different imaging conditions. Experimental results have demonstrated the greater effectiveness of UDA-based models compared to traditional supervised models in reconstructing the BE area, an early cancer precursor [32].

### 3.4. Domain Adaptation for Alzheimer’s Disease Classification

Magnetic resonance imaging is an excellent diagnostic technique. When using computer-aided diagnosis to diagnose dementia, characterizing of the brain anatomy is promising for diagnosing and classifying Alzheimer’s disease (AD), mild cognitive impairment, and normal controls. Large, multicenter datasets are available for studying AD and supporting the training of complex classification models. Interestingly, one study using this classification model found that all the participating groups overestimated the accuracy of their method. One of the main reasons for this reduced classification accuracy was the variation in the distribution between the training and test data. Such problems involve domain adaptation, in which the model is taught using a source dataset and transferred to a target dataset with different properties based on instance weighting. They used supervised domain adaptation, in which the source domain was the training domain with labeled data, whereas the target domain was the test domain with only a fraction of labeled data. This could be a possible AD classification method for improving recognition rate, regardless of the data source [33,34,35].

### 3.5. Semi-Supervised Learning with GANs for Chest X-ray Classification with the Ability of Data Domain Adaptation

Because of privacy laws, medical industry standards, and a lack of integration into medical information systems, the sources of medical imaging data are not as rich as those of other fields in computer vision. Therefore, it is a challenge to develop deep learning algorithms for medical imaging. Even if data are available, unstructured or inadequate labeling becomes an obstacle to utilizing these data. To address this problem, medical images may be annotated; however, this is a time-consuming and costly process. As a result, when deep-learned classifiers are trained on a particular training dataset and then tested in production on data from a different domain source, their performance and accuracy are determined. Madani et al. addressed the problems of labeled data scarcity and data domain differences using GANs. They confirmed that deep GANs can learn the visual structures of medical imaging domain sources (particularly chest X-rays). They proposed a semi-supervised architecture for GANs, which is capable of learning from both labeled and unlabeled images. Their results showed that, when labeled data are limited, a semi-supervised GAN-based network requires one-order-of-magnitude-less labeled training data to achieve a performance comparable with that of a supervised CNN classifier. In other words, a performance similar to that of supervised training techniques is achieved with a considerably reduced annotation effort. They attributed this result to GANs being able to learn the structures in unlabeled data when using unsupervised learning methods, which significantly offsets the low number of labeled data samples [36,37,38].

### 3.6. UDA-Based Coronavirus Disease of 2019 (COVID-19) Infection Segmentation Network

The automatic segmentation of infected lung regions in computed tomography (CT) images has proven to be an effective diagnostic tool for COVID-19. However, because of the limited number of pixel-level labeled medical images, accurate segmentation remains a major challenge. More recently, authors generated synthetic COVID-19 CT data to promote the computer-aided diagnostic ability for COVID-19, making it possible to train deep models on synthetic images and computer-generated annotations. However, this study also found that a model directly trained on synthetic data may fail to produce accurate results for real COVID-19 CT images, because of domain shift. To resolve this problem, the authors proposed a UDA-based segmentation network to improve the segmentation performance for the infected areas in the COVID-19 CT images. They proposed making full use of synthetic data and limited unlabeled real COVID-19 CT images to train the segmentation network jointly in order to introduce a richer diversity. This approach reduced domain shift by forcing the features from different domains to fool the discriminator, leading to features from different domains exhibiting a similar distribution. This method played an important role in diagnosing COVID-19 by quantifying the infected areas of the lungs. Domain adaptation also demonstrated positive effects on medical image segmentation [39,40].

**Table 1 diagnostics-13-03023-t001:** Example of studies on application of domain adaptation in the medical images analysis.

Reference	Medical Instrument	Task	Module	Result
Laiz et al. (2019) [25]	Capsule endoscopy	Improve the generalization of a model over different datasets from different versions of WCE hardware.	Deep metric learning, based on the triplet loss function	Just a few labeled images from a newer camera set, a model that has been trained with images from older systems can be easily adapted to the new environment.
Zhan et al. (2020) [28]	Colonoscopy	Colon polyp detection, images to real-time videos	Ivy-Net	Ivy-Net to alleviate the domain gap between colonoscopy images from historical medical reports and real-time videos.
Celik et al. (2012) [32]	Gastroscopy	Barret’s esophagus area segmentation	UDA	UDA method generalizes on different imaging modalities showing improved segmentation accuracy.
Wachinger et al. (2016) [35]	MRI	Alzheimer’s disease classification	Supervised domain adaptation(SDA)	Domain adaptation with instance weighting yields the best classification results
Madani et al. (2018) [38]	Chest X-ray	Abnormality detection	GANs	Annotation effort is reduced to achieve similar performance through supervised training techniques.
Chen et al. (2021) [39]	CT	Automatic segmentation of infection area	UDA	Segmentation network to learn the domain-invariant feature, so that the robust feature can be used for segmentation.

## 4. Perspective and Future Direction

The latest AI models for automated GI endoscopy reading have fast learning speeds and fairly high reading accuracies. Thus, they have a high potential for clinical use. However, these AI models have several shortcomings and exhibit significant performance degradation when the data format or domain is slightly different from the training data format. This issue is called a “domain shift problem”. This is one of the major factors preventing the use of AIs at the preclinical level. This domain shift problem has been observed in GI endoscopy and other medical imaging processes, because gathered medical images are obtained using different scanners with different scanning parameters and involve different subject cohorts. As a result, and as mentioned above, heterogeneity among medical image datasets is inevitable.

For this reason, domain adaptation techniques have gained attention among developers. A domain is related to a specific dataset’s feature space and features a marginal probability distribution [41]. Originally, AI engineers have adopted domain adaptation techniques to enable efficient algorithm learning by creating new virtual data and extracting the main features of obtained images. However, image shifts through domain adaptation can minimize the distribution gap among different but partially related domains in medical image analyses. For domain adaptation, it is assumed that the domain feature spaces and tasks remain the same, but there is a difference in the marginal distributions between the source and target domains. Many applications of domain adaptation in GI endoscopy and medical image analyses have been reported. Laiz et al. first proposed the concept of triplet loss function for domain adaptation in capsule endoscopy, and reported that it could improve the reading accuracy by improving the generalization of the datasets obtained from different systems [25]. Zhan et al. improved the polyp detection in colonoscopy videos by using Ivy-Net to bridge the domain gap between colonoscopy images and real-time videos [28]. Celik et al. applied a UDA framework for the segmentation of BE, which is a precancerous lesion [32]. Wachinger et al. used AD classification to improve the recognition rate, regardless of the data source, using supervised domain adaptation, in which the source domain was the training domain with labeled data [35]. Madani et al. confirmed the positive effects of domain adaptation on chest X-ray classification [38]. Chen et al. developed a novel domain adaptation module that could achieve a good segmentation performance on COVID-19 CT images, even when no annotations were provided [39].

Many researchers have used only supervised learning for machine learning strategies in AI studies on medical images. Active learning and unsupervised learning are still not available because of their low outcome accuracy. It takes a lot of time to collect, classify, and label the learning materials for an AI algorithm’s development. If domain adaptation could be applied to new medical images, the previously developed algorithm would be able to bring about a robust accuracy to those images. Domain adaptation could be also applied to medical images obtained using old medical devices. Of course, it is very useful when AI learning materials are lacking.

Can AI-assisted interpretation replace the current time-consuming conventional reading in real clinical fields? In terms of the immediate future, our answer is no. It may be some time before such a situation comes to pass. Although many published studies have shown the very high sensitivity and specificity of AI algorithms in the interpretation of medical images, it should be noted that the training and testing sets in these studies were designed for selected images, not medical images obtained from real-world clinical environments. 

Actual medical images are images in which various lesions, non-specific findings, normal variants, blurring, and images obtained from different hospitals are mixed. These are different from the testing sets that are used by researchers. The reason for the low accuracy of AI is that too many false positives are produced in real-world applications. Although the test results are usually very good in limited conditions, when compared to external results from other institutes, a low accuracy is typically reported, indicating suboptimal results. The problem of overfitting is important in this regard. To overcome the overfitting problem, domain adaptation is an engineering technique that helps AI to be actually and universally used in real clinical fields. We would like to emphasize that the availability of good data is still the first most important point in AI studies. Large amounts of data are important, as is an even distribution of the various different images constituting the training data set. For good data, a reference standard should be established by consensus from a panel of experienced professionals.

Domain adaptation has the potential to be utilized in various fields of medical image analysis. In addition, the need to label a large-scale dataset can be alleviated by the use of a virtually generated image using domain adaptation. By such means, a time-consuming, laborious, and expensive task can be avoided. Moreover, this technique is expected to improve the accuracy of AI reading, while solving the problem of heterogeneous image distribution in multicenter studies. Ultimately, the AI model developed for lesion detection in GI endoscopy and other medical fields can easily be utilized by domain adaptation.

## 5. Conclusions

Automated lesion detection through AI can be seen as a promising field of research. However, as images with different characteristics are extracted, depending on the device manufacturer and the imaging environment, the so-called “domain shift problem” occurs, in which the developed AI has a poor versatility. Domain adaptation is a tool that can generate a new, converted image suitable for other domains. This is a promising tool for reducing the appearance differences among images collected from different devices and environments. It is expected to improve the AI reading accuracy for heterogenous image distribution in medical image analyses.

## Figures and Tables

**Figure 1 diagnostics-13-03023-f001:**
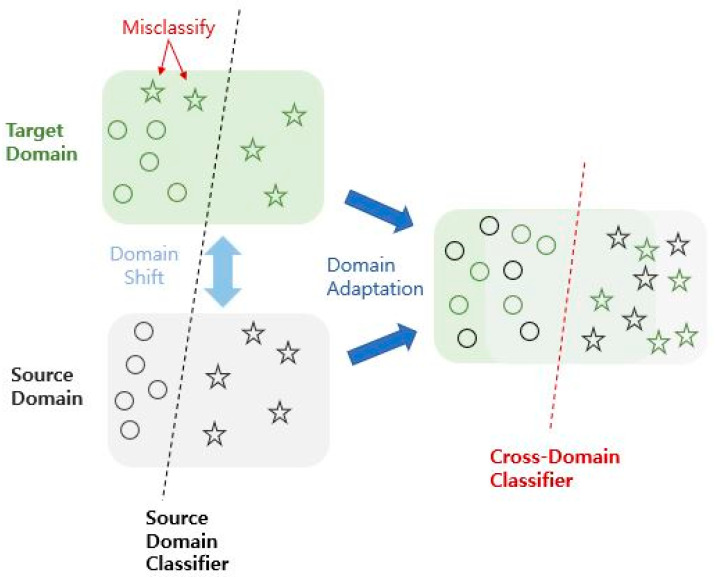
Occurrence of domain shift and process of domain adaptation.

**Figure 2 diagnostics-13-03023-f002:**
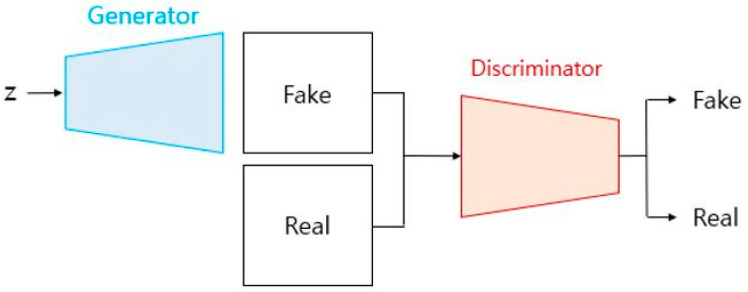
Principle of generative adversarial network (GAN).

**Figure 3 diagnostics-13-03023-f003:**
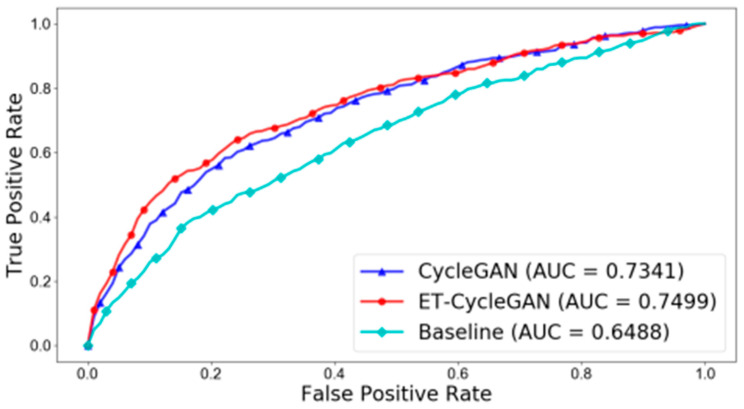
Changes in the receiver operating characteristic curves of artificial intelligence after domain adaptation.

**Figure 4 diagnostics-13-03023-f004:**
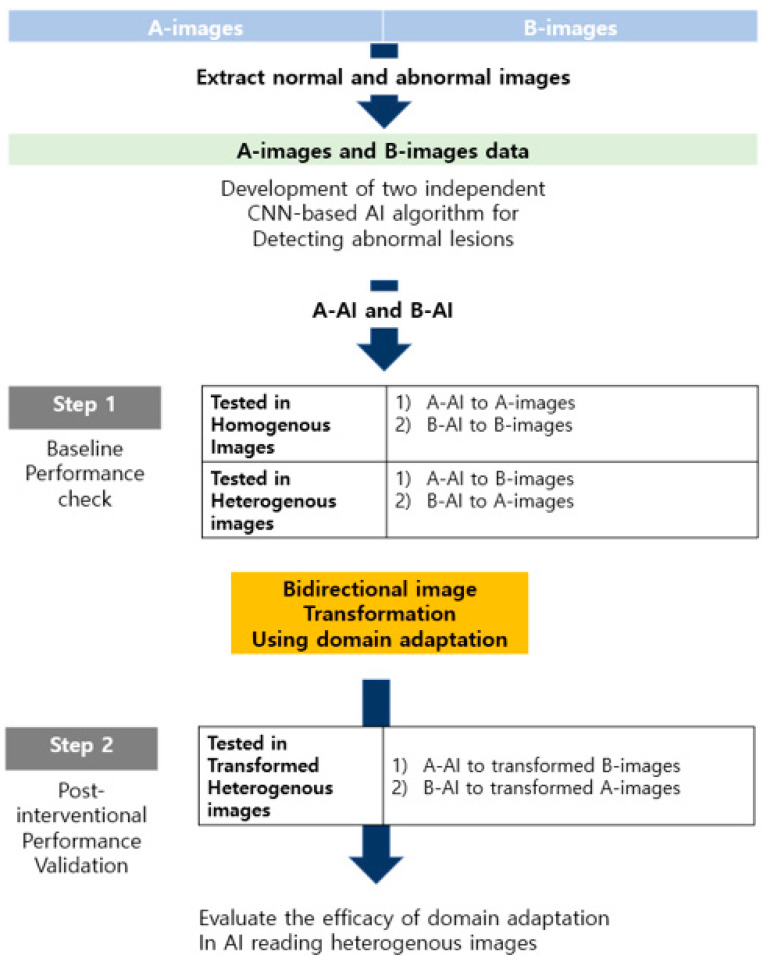
Example: application of domain adaptation in capsule endoscopy.

## Data Availability

Not applicable.

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
