# Peer review of "The Advent of Domain Adaptation into Artificial Intelligence for Gastrointestinal Endoscopy and Medical Imaging"

_diagnostics, 2023, doi:10.3390/diagnostics13193023_

Round 1

Reviewer 1 Report

The following points should be addressed although the manuscript is not well organized.

1. The abstract must be rewritten.

2. The title should be eye catching.

3. The introduction should start with an overview of the digestive system, its benefits, causes of diseases and its treatment, types of diagnostic techniques, deficiencies in manual diagnosis, and the importance of artificial intelligence to address deficiencies in manual diagnosis.

4. The title is about digestive diseases while the next point is about Alzheimer's disease. Why

"Domain adaptation for Alzheimer's disease classification."

5. The title is about gastroenterology, while the next point is about chest x-ray. Why?

"Semi-supervised learning with GANs for chest x-ray classification with the ability of data domain adaptation"

6. The heading is about gastroenterology while the next point is about COVID-19. Why?

"UDA-based coronavirus disease of 2019 (COVID-19) infection segmentation network."

7. A methodology for the research should be drawn figure so that the reader can easily follow the progress of the work.

8. Finally, the manuscript is not well organized.

Minor editing of English language required.

Author Response

Thank you for your all kind comments. There are not many studies applied to medicine regarding the domain adaptation, which is especially rare in the field of GI endoscopy. We have introduced published examples to help readers understand the domain adaptation as much as possible. Please understand our intention.

Reviewer 2 Report

The authors proposed a study in the Domain adaptation for gastrointestinal endoscopy. my observations are:

1.The formatting of the article is poor and need restructuring

2. The novelty and outcomes should be clear in the abstract,

3. The article is a kind of review article so authors need to mention sub scetion numbering

4. cite some more recent works as well

5. A comparasion with recent methods need to discuss

6. SOme of the sentences are errorous and need grammatical revision.

7. Add paper classification as the last paragraph in the introduction section.

8. At some places the author wrote “we” suggested while the article is a kind of study. Need to justify. If so then what is the method the author suggested.

9. Figure 3 seems to be irrelevant as far the study is concerned. There is no such figure exist which shows the flow of proposed work

10. Need to add and compare the recent works as part of this study.

need revision

Author Response

Thanks for your all good comments. Our research group tried various engineering methods to increase AI accuracy. We conducted the research on the domain adaptation of capsule endoscopy. FIGURE 3 introduced our design but unfortunately didn't publish the domain adaptation of capsule endoscopy due to industry's interest conflicts.  We changed the arrangement of figure 3 and figure 4 and supplemented the contents to explain the figure 3.    

Reviewer 3 Report

Congratulation for a very interesting review in a crucial topic regarding the applicability of AI models to heterogenous medical datasets and medical conditions fully deserving publication. 

I dont feel qualified to fully  judge the quality of english

Author Response

Manuscript ID: 2567944

Manuscript title: Advent of Domain Adaptation into Artificial Intelligence of Gastrointestinal Endoscopy and Medical Imaging

The following points should be addressed although the manuscript is not well organized.

1.The abstract must be rewritten.

 Thank you for your kind comments. We modified that the abstract to include the contents of the text more implicitly.

After revision : In this paper, we review the history and basic characteristics of domain shift and domain adaptation. We also address its use in gastrointestinal endoscopy and the medical field more generally through published examples and perspectives and future direction.

  1. The title should be eye catching.

 Thank you for your kind comments. We are sorry to missing that our manuscript include not only domain adaptation in gastrointestinal endoscopy but also other medical fields. We also think more specific title is needed. We change the title from ‘Domain Adaptation for Gastrointestinal Endoscopy’ to ‘Advent of Domain Adaptation into Artificial Intelligence of Gastrointestinal Endoscopy and Medical Imaging’.

  1. The introduction should start with an overview of the digestive system, its benefits, causes of diseases and its treatment, types of diagnostic techniques, deficiencies in manual diagnosis, and the importance of artificial intelligence to address deficiencies in manual diagnosis.

Thank you for your kind comments. We further describe the role of endoscopy and the goal of AI research in the text.

After revesion : Gastrointestinal (GI) endoscopy is an active area of AI research. Upper endoscopy, colonoscopy and capsule endoscopy detect the inflammation, bleeding foci, preneoplastic lesion and gastrointestinal cancer. Goals of AI research are to improve the ability to detect the abnormal lesions as well as enhance the gastrointestinal imaging, quality control and clinical efficiency in the real clinical environment.

  1. The title is about digestive diseases while the next point is about Alzheimer's disease. Why "Domain adaptation for Alzheimer's disease classification." 5. The title is about gastroenterology, while the next point is about chest x-ray. Why? "Semi-supervised learning with GANs for chest x-ray classification with the ability of data domain adaptation" 6. The heading is about gastroenterology while the next point is about COVID-19. Why? "UDA-based coronavirus disease of 2019 (COVID-19) infection segmentation network."

4-6 : Thank you for your important comments.There are not many studies applied to medical fields regarding the domain adaptation, which is especially rare in the field of GI endoscopy. We have introduced published examples to help readers understand the domain adaptation as much as possible. Please understand our intention.

  1. A methodology for the research should be drawn figure so that the reader can easily follow the progress of the work. 

Thank you for your kind comments. Our research group tried various engineering methods to increase AI accuracy. We conducted the research on the domain adaptation of capsule endoscopy FIGURE 3 introduced our design but unfortunately didn't publish the domain adaptation of capsule endoscopy due to industry's interest conflicts. We changed the arrangement of figure 3 and figure 4 and supplemented the contents to explain the figure 3. And we change the content of the research we conducted has been reordered to make it easier for readers to understand.

After revesion : Figure 3. Changes in the receiver operating characteristic curves of artificial intelligence after domain adaptation.

Figure 4. Example: Application of domain adaptation in capsule endoscopy

We recently conducted a domain adaptation study using WCE. The latest AI models for automated WCE reading are useful with respect to reductions in reading time. In addition, we used AI for automated abnormal-lesion detection in small-bowel WCE and found that the performance of AI-assisted interpretation was comparable to that of experienced endoscopists for abnormal-lesion detection without improvement. However, the performance of the optimized WCE AI algorithm deteriorated when applied to data from other hospitals. In addition, my previous study showed that a binary classification model (two categories of images: clinically insignificant images including normal mucosa, bile, air bubbles and debris; and significant lesion images including inflammation, abnormal vascular lesion and bleeding) produced excellent outcomes in an internal hospital test, with high sensitivity even in unseen images. Unfortunately, this model showed subclinical outcomes in an external test at third-party hospital. This phenomenon is also commonly experienced in AI application of other images in real-world situations. This constitutes a major obstacle to the universal use of AI in medical fields. The domain shift problem has been also observed in WCE. AI engineers have adopted the domain adaptation technique to enable efficient algorithm learning by creating new virtual data and extracting main features of a collected image. An A-image and a B-image were handled separately and divided into training and validation (test) sets. Two independent AI algorithms (A-AI and B-AI) were built using the training images from the two groups. These were trained to discriminate between normal and significantly abnormal images. After training, each AI algorithm was applied to a homogeneous test set. These algorithms were then tested using a heterogeneous test set (e.g., applying A-AI to the B-image test set, or applying B-AI to the A-image test set) to determine whether the performance improved or deteriorated. Their performance was evaluated using heterogeneous test images after several up-to-date domain adaptation techniques were applied to the test images. Thus, we attempted to evaluate the extent to which the domain adaptation technique could improve the reading accuracy in heterogeneous images (Figure 4). We tested three different domain adaptation architectures introduced from 2017 to the present: CycleGAN, discoGAN, and MUNIT. These domain adaptation methods enable original images of WCE to be transformed and reconstructed into a new image to match distribution of the target domain. This process allows the newly created image to possess a style similar to the heterogenous capsule image. We also compared various domain adaptation techniques for an EfficientNet-based algorithm and a ResNet-based algorithm. We found that domain adaptation was effective regardless of algorithms. In addition, domain adaptation using CycleGAN resulted in an additional AI performance improvement. A similar improvement in performance improvement was obtained using discoGAN. The improvement in efficacy outcome was the greatest when domain adaptation was performed with CycleGAN.

  1. Finally, the manuscript is not well organized.

Thank you for your kind comments. We closely checked the organization of the manuscript one more time.

Manuscript ID: 2567944

Manuscript title: Domain adaptation for gastrointestinal endoscopy

Comments and Suggestions for Authors

The authors proposed a study in the Domain adaptation for gastrointestinal endoscopy. my observations are:

1.The formatting of the article is poor and need restructuring

Thank you for your kind comments. We closely checked the organization of the manuscript one more time.

  1. The novelty and outcomes should be clear in the abstract,

Thank you for your kind comments. We modified that the abstract to include the contents of the text more implicitly.

After revision : In this paper, we review the history and basic characteristics of domain shift and domain adaptation. We also address its use in gastrointestinal endoscopy and the medical field more generally through published examples and perspectives and future direction.

  1. The article is a kind of review article so authors need to mention sub scetion numbering

Thank you for your important comments. We numbered each section and sub section.

After revesion :

  1. Introduction
  2. Fundamentals and history of domain adaptation and domain shift
  3. Application of domain adaptation GI endoscopy and medical field
    1
    Using the triplet loss for domain adaptation in wireless capsule endoscopy (WCE)

3.2 Colonoscopy polyp detection: domain adaptation from medical report images to                     real-time videos

3.3 Unsupervised adversarial domain adaptation for Barrett’s segmentation

3.4 Domain adaptation for Alzheimer’s disease classification

3.5 Semi-supervised learning with GANs for chest X-ray classification with the ability of data  domain adaptation

3.6 UDA-based coronavirus disease of 2019 (COVID-19) infection segmentation network

  1. Perspective and future direction
  2. Conclusion

  1. cite some more recent works as well

Thank you for your important comments. We try to cite studies within the last 5 years. And we reviewed some of the latest papers and added them to our reference.

Sumiyama, K., et al., Artificial intelligence in endoscopy: Present and future perspectives. Dig Endosc, 2021. 33(2): p. 218-230.

Chen, H., et al., Unsupervised domain adaptation based COVID-19 CT infection segmentation network. Appl Intell (Dordr), 2022. 52(6): p. 6340-6353.

Dumoulin, F.L., et al., Artificial Intelligence in the Management of Barrett's Esophagus and Early Esophageal Adenocarcinoma. Cancers (Basel), 2022. 14(8).

Kim, S.H. and Y.J. Lim, Artificial Intelligence in Capsule Endoscopy: A Practical Guide to Its Past and Future Challenges. Diagnostics (Basel), 2021. 11(9).

Nam, J.H., K.H. Lee, and Y.J. Lim, Examination of Entire Gastrointestinal Tract: A Perspective of Mouth to Anus (M2A) Capsule Endoscopy. Diagnostics (Basel), 2021. 11(8).

Yu, M., et al., Domain-Prior-Induced Structural MRI Adaptation for Clinical Progression Prediction of Subjective Cognitive Decline. Med Image Comput Comput Assist Interv, 2022. 13431: p. 24-33.

Xu, G.X., et al., Cross-Site Severity Assessment of COVID-19 From CT Images via Domain Adaptation. IEEE Trans Med Imaging, 2022. 41(1): p. 88-102.

  1. A comparasion with recent methods need to discuss

Thank you for your kind comments. We try to described at Table 1 summarizes the domain adaptation techniques of the last five years.

  1. Some of the sentences are errorous and need grammatical revision.

Thank you for your kind comments. We received english editing once again and please check the attached certificate.

  1. Add paper classification as the last paragraph in the introduction section.

Thank you for your important comments. We numbered each section.

  1. Introduction
  2. Fundamentals and history of domain adaptation and domain shift
  3. Application of domain adaptation GI endoscopy and medical field
    1
    Using the triplet loss for domain adaptation in wireless capsule endoscopy (WCE)

3.2 Colonoscopy polyp detection: domain adaptation from medical report images to                     real-time videos

3.3 Unsupervised adversarial domain adaptation for Barrett’s segmentation

3.4 Domain adaptation for Alzheimer’s disease classification

3.5 Semi-supervised learning with GANs for chest X-ray classification with the ability of data  domain adaptation

3.6 UDA-based coronavirus disease of 2019 (COVID-19) infection segmentation network

  1. Perspective and future direction
  2. Conclusion

  1. At some places the author wrote “we” suggested while the article is a kind of study. Need to justify. If so then what is the method the author suggested.

Thank you for your kind comments. We are sorry that there was some confusing description in our manuscript. In manuscrip, the sentence with "We" was carefully examined and replaced with "Many researchers".

After revision : Many researchers used only supervised learning for machine learning strategies in AI study of medical images.

  1. Figure 3 seems to be irrelevant as far the study is concerned. There is no such figure exist which shows the flow of proposed work

Thank you for your important comments. Our research group tried various engineering methods to increase AI accuracy. We conducted the research on the domain adaptation of capsule endoscopy FIGURE 3 introduced our design but unfortunately didn't publish the domain adaptation of capsule endoscopy due to industry's interest conflicts.  We changed the arrangement of figure 3 and figure 4 and supplemented the contents to explain the figure 3.    

  1. Need to add and compare the recent works as part of this study.

 Thank you for your kind comments. We try to described at Table 1 summarizes the domain adaptation techniques of the last five years. And we reviewed some of the latest papers and added them to our reference.

Table1

Sumiyama, K., et al., Artificial intelligence in endoscopy: Present and future perspectives. Dig Endosc, 2021. 33(2): p. 218-230.

Chen, H., et al., Unsupervised domain adaptation based COVID-19 CT infection segmentation network. Appl Intell (Dordr), 2022. 52(6): p. 6340-6353.

Dumoulin, F.L., et al., Artificial Intelligence in the Management of Barrett's Esophagus and Early Esophageal Adenocarcinoma. Cancers (Basel), 2022. 14(8).

Kim, S.H. and Y.J. Lim, Artificial Intelligence in Capsule Endoscopy: A Practical Guide to Its Past and Future Challenges. Diagnostics (Basel), 2021. 11(9).

Nam, J.H., K.H. Lee, and Y.J. Lim, Examination of Entire Gastrointestinal Tract: A Perspective of Mouth to Anus (M2A) Capsule Endoscopy. Diagnostics (Basel), 2021. 11(8).

Round 2

Reviewer 1 Report

 Accept in present form

 Minor editing of English language required